# Evaluation of Open-Source Tools for Differential Privacy

**DOI:** 10.3390/s23146509

**Published:** 2023-07-19

**Authors:** Shiliang Zhang, Anton Hagermalm, Sanjin Slavnic, Elad Michael Schiller, Magnus Almgren

**Affiliations:** 1Computer Science and Engineering, Chalmers University of Technology, SE-41296 Gothenburg, Sweden; shiliang@chalmers.se (S.Z.); anthage@student.chalmers.se (A.H.); slavnic@student.chalmers.se (S.S.); almgren@chalmers.se (M.A.); 2Department of Informatics, University of Oslo, NO-0373 Oslo, Norway

**Keywords:** differential privacy, open-source tools, evaluation

## Abstract

Differential privacy (DP) defines privacy protection by promising quantified indistinguishability between individuals who consent to share their privacy-sensitive information and those who do not. DP aims to deliver this promise by including well-crafted elements of random noise in the published data, and thus there is an inherent tradeoff between the degree of privacy protection and the ability to utilize the protected data. Currently, several open-source tools have been proposed for DP provision. To the best of our knowledge, there is no comprehensive study for comparing these open-source tools with respect to their ability to balance DP’s inherent tradeoff as well as the use of system resources. This work proposes an open-source evaluation framework for privacy protection solutions and offers evaluation for OpenDP Smartnoise, Google DP, PyTorch Opacus, Tensorflow Privacy, and Diffprivlib. In addition to studying their ability to balance the above tradeoff, we consider discrete and continuous attributes by quantifying their performance under different data sizes. Our results reveal several patterns that developers should have in mind when selecting tools under different application needs and criteria. This evaluation survey can be the basis for an improved selection of open-source DP tools and quicker adaptation of DP.

## 1. Introduction

Privacy relates to one’s ability to decide on the manner, context, and timing that one’s personal information is managed by others. Data privacy protection is important for guaranteeing human dignity, safety, and self-determination as well as for guarding proprietary rights and economic interests. Despite the extensive academic research and legal recognition that data privacy protection has received, e.g., General Data Protection Regulation (GDPR) [1], Health Insurance Portability and Accountability Act (HIPAA) [2], and California Consumer Privacy Act (CCPA) [3], the practice often does not use quantified methods for protecting data privacy. Differential privacy (DP) is a prominent proposal for quantifying privacy protection. It promises quantified indistinguishability between individuals who consent to share their privacy-sensitive information and those who do not. DP aims at delivering this promise by including well-crafted elements of random noise in the published data, and thus there is an inherent tradeoff between the degree of privacy protection and the ability to utilize the protected data. Recently, several open-source tools have been proposed for DP provision. To the best of our knowledge, there is no comprehensive study for comparing these open-source tools with respect to their ability to balance DP’s inherent tradeoffs as well as the use of system resources. This work proposes an open-source evaluation framework for DP tools and services. Using this framework, we offer evaluation results for OpenDP Smartnoise, Google DP, PyTorch Opacus, Tensorflow Privacy, and Diffprivlib. We consider both discrete and continuous attributes by quantifying their performance under different data sizes. Our results reveal several patterns that users should have in mind when selecting DP tools.

Concretely, in this paper, we evaluate open-source differential privacy (DP) tools and services for protecting privacy-sensitive information in data systems. We develop a comprehensive evaluation framework and provide insights into the performance of these tools for various data analysis tasks. Our work facilitates the selection of DP tools by developers and aims to bridge the gap between theoretical and applied research on DP. In the rest of this section, we articulate the definition of differential privacy (Section 1.1). Next, we review the related open-source tools (Section 1.2) before delving into our evaluation approach (Section 1.3). We then proceed to stating the specific way in which we advance the state of the art (Section 1.4).

### 1.1. Differential Privacy

Dwork proposed differential privacy (DP) [4] as a means for guaranteeing that all individuals will be exposed to essentially the same risk of jeopardizing their privacy. This is completed by quantifying the probability of his or her privacy-sensitive information being included in a DP analysis. That is, the analysis ensures that privacy-sensative information cannot be effectively revealed, regardless of the adversary’s computational power or access to any additional information that may exist, or will ever exist.

The ϵ-DP definition is one of the formulations of DP, where ϵ is the key parameter. Specifically, ϵ quantifies the privacy loss or the amount of noise added to the computation to achieve privacy guarantees. A smaller value of ϵ indicates a stronger privacy guarantee. We describe the definition of ϵ-DP protection, which considers a probabilistic program (or mechanism) *M*; see [5] for details. The program *M* is said to provide ϵ-DP protection for all events *S*, S⊆range(M), given two datasets *D* and $D^{\prime }$, such that $\mathrm{Pr}\left[M\left(D\right)\in S\right]\leq e^{\epsilon }\cdot Pr\left[M\left(D^{\prime }\right)\in S\right]+\delta$, where range(M) denotes the output range with a given input, and Pr[] denotes probability distribution and δ≥0 is a predefined constant that quantifies the probability of an additional privacy breach beyond the specified privacy guarantee. The parameter ϵ refers to the privacy budget, which controls the level of privacy guarantee achieved by mechanism *M*. In other words, DP guarantees that the addition or removal of information related to a single individual in a dataset essentially does not affect the result of any analysis or query and limits the risk of privacy disclosing associated with providing or refraining from providing (privacy-sensitive) information to a dataset.

### 1.2. A Brief Review of Open-Source Tools with DP Services

We briefly review the most relevant open-source DP services. In this work, the term statistical queries refers to the analysis of data to extract statistical features, e.g.,  the operation of SUM, AVERAGE, COUNT, and HISTOGRAM.

#### 1.2.1. OpenDP Smartnoise

OpenDP Smartnoise [6] has its roots in Harvard University Privacy Tools Project. This project gained experience in building and deploying PSI [7], a system developed to share and explore privacy-sensitive datasets with privacy protections of DP, and ultimately contributed to their efforts toward Smartnoise. OpenDP Smartnoise has also incorporated insights from other DP tools, such as PinQ [8], ϵktelo [9], PrivateSQL [10], Fuzz [11], and LightDP [12]. While most of the tools like PSI and PinQ are research prototypes, OpenDP Smartnoise is now putting efforts into further developing DP concepts into production-ready tools. With those features, OpenDP Smartnoise has obtained significant popularity from the developer community, with more than two hundred stars in its open-source software repository.

#### 1.2.2. Google DP

Google released an open-source version of its DP library that empowers some of its core products [13]. Available in Java and Go, this library captures years of Google’s developer experience and offers practitioners and organizations potential benefits from its implementation, with a fairly low entrance level of expertise in DP.

#### 1.2.3. Opacus

PyTorch Opacus [14] is Facebook’s DP library for machine learning (ML) services, built on top of PyTorch. It is developed in collaboration with Facebook AI Research, the PyTorch team, and OpenMined, an open-source community dedicated to developing privacy techniques for ML and AI. The service from Opacus targets both ML practitioners and professional DP researchers with its general and specific features.

#### 1.2.4. Tensorflow Privacy

TensorFlow Privacy [15] (TFP) is an ML framework developed and released by Google, initially inspired by the work of Abadi et al. [16], who implemented a similar optimizer for TensorFlow and a privacy cost tracker. It emerges and adapts DP mechanisms to TensorFlow to allow users to leverage differential privacy in the training of ML models. Furthermore, TFP is configurable, and developers can define their own ML models, with which developers can implement their operators in their applications. With its flexibility and DP services, TFP has become an open DP tool that is leveraged and contributed to by a large developer community.

#### 1.2.5. Diffprivlib

Developed by the industrial giant IBM, Diffprivlib [17] empowers differential privacy in machine learning tasks, including classification, regression, clustering, dimensionality reduction, and data regularization. It is supposed to be a general-purpose tool for conducting experiments, investigations, and application developments with differential privacy. With its detailed product manual (https://diffprivlib.readthedocs.io/en/latest/, accessed on 30 May 2023), practitioners of different levels can easily find what they need during their interaction with Diffprivlib. As a result, Diffprivlib’s open repository [18] has gained considerable attention amongst developers.

#### 1.2.6. Chorus

Chorus [19,20] utilizes a cooperative architecture to achieve DP statistical queries. It leverages industrial-grade database management systems (DBMS) for data processing tasks and even queries that need to be modified or, in some cases, entirely rewritten. This architecture has three primary components, namely rewriting, analysis, and post-processing. The rewriting component is used to modify queries to perform functions like clipping, the analysis component to analyze queries to determine different properties such as required noise to satisfy differential privacy, and the post-processing component to process the result of the queries. An example from Chorus is the implementation of a summation mechanism with clipping. The rewriting component can modify the original query so that the DBMS executes the clipping and the summation, leaving the rest of the summation mechanism to the analysis and the post-processing component.

What separates Chorus from previous work is that it is DBMS-independent. Unlike an integrated approach, Chorus does not require modifying or affecting the database or changing to a purpose-built database engine. Therefore, Chorus can leverage DMBS to ensure scalability when working on datasets that hold large amounts of data.

Added safeguards can be necessary when deploying Chorus to minimize the chance of a malicious actor acquiring sensitive data. For example, in the case of Chorus’ deployment on Uber, privacy-sensitive data were only available through a centralized query interface, which was protected along with the privacy budget account and the DMBS from tampering.

While the early repository of Chorus archived by Uber is deprecated (https://github.com/uber-archive/sql-differential-privacy, accessed on 30 May 2023), a new version emerged that is maintained and active among the open-source community (https://github.com/uvm-plaid/chorus, accessed on 30 May 2023). The new repository of Chorus has not gained much attention since its relatively recent release. Nevertheless, Chorus showed strength in its early version, and, thus, more concrete results and popularity among developers can be expected for its current version. In this work, we focus on the evaluation of DP tools that have already gained significant popularity; see Section 1.3.

### 1.3. Evaluation Approach

Several open-source tools are available for applying DP in statistical queries [21] and machine learning [17]. In this paper, we refer to statistical queries as the retrieval of features and aggregated information from datasets, e.g., SUM, COUNT, and AVERAGE. In machine learning tasks, models are constructed using training data from a given dataset. For example, linear regression is an effective linear modeling approach as it captures the relationship between a scalar response and one or more explanatory variables. Those two kinds of tasks, i.e., statistical query or machine learning, are prevailing tools for application development, and, thus, we select them to serve as evaluation test cases. Our evaluation strategy is summarized in Table 1.

Our choice regarding the studied DP tools was based on the support that each of these tools received from comparably larger communities, technology companies, and research institutions as well as their wide acceptance in the open-source community. We also required a sufficiently long history of being free from known bugs or fixable bugs [22], as well as developer-friendly documentation. Our evaluation criteria for the studied DP tools focus on data utility and system overhead, i.e., running time and required memory. The considered tools are evaluated within two task domains, i.e.,  statistical query and machine learning. Since tools of different categories follow different evaluating procedures, we analyze the evaluation results within every single domain. During the evaluation, we adopt a diversity of settings to investigate how different tools’ performances vary under different conditions.

We use the open-source data of United States Health Reform Monitoring Survey [23] and the UCI Parkinson dataset [24] for experiments in statistical queries and machine learning. We carefully selected these datasets since they fulfill two criteria: alignment with the functionalities of the investigated tools and diversified data settings. Our chosen datasets cover both statistical analysis and machine learning tasks supported by the considered tools. For example, the Health Survey dataset includes categorical integer data, while the other involves continuous data, ensuring a comprehensive evaluation across different data types. This deliberate diversity offers valuable insights into how the tools perform with varied data.

#### 1.3.1. The United States Health Reform Monitoring Survey Data

The Massachusetts Health Reform Survey (MHRS) is an annual telephone survey conducted to track the impacts of Massachusetts’ comprehensive healthcare reform bill, which aimed to achieve near universal coverage in the state. The survey was initiated in 2006, just before the implementation of key elements of the law, and has been conducted in various years since then, including 2006–2010, 2012, 2013, 2015, and 2018. The survey sample consists of a random selection of working-age adults in Massachusetts each year. The dataset size varies by year, with sample sizes ranging from approximately 2000 to 4000 non-elderly adults. The survey covers various aspects related to healthcare, including insurance status, access to and utilization of healthcare services, out-of-pocket costs, medical debt, insurance premiums, covered services, and health and disability status. The survey questions are primarily drawn from established and validated surveys, with some modifications to address specific concerns in Massachusetts. Over time, questions have been added or removed to address emerging issues and maintain a reasonable survey length.

#### 1.3.2. The UCI Parkinson Dataset

This dataset is focused on tracking the progression of Parkinson’s disease (PD) symptoms using speech tests. It introduces a method for remote assessment of the unified Parkinson’s disease rating scale (UPDRS) through self-administered speech tests, providing clinically useful accuracy. The dataset is consdired to be the largest PD speech database in existence, consisting of approximately 6000 recordings from 42 PD patients who participated in a six-month multicenter trial. The dataset contains speech recordings obtained from PD patients. The recordings include sustained vowel phonations, where the patients are asked to hold the frequency of phonation steady for as long as possible. The recorded voice signals are analyzed using signal processing algorithms to extract clinically useful features that reflect the progression of PD symptoms. The dataset also includes corresponding UPDRS scores, which serve as the metric to assess the presence and severity of symptoms. The dataset’s main purpose is to demonstrate the feasibility of frequent, remote, and accurate UPDRS tracking using speech tests. By mapping the extracted speech features to UPDRS scores through regression techniques, the study aims to provide objective tools for assessing PD and enable telemonitoring frameworks for large-scale clinical trials of novel PD treatments.

### 1.4. Our Contribution

We study a critical aspect of data systems, which is the protection of privacy-sensitive information. Our study evaluates open-source differential privacy (DP) tools and services. The area of privacy protection has a well-recorded history of failed solutions. DP is a leading framework that offers qualitative guarantees for the protection of privacy-sensitive information. The implementation of tools for providing DP solutions has its own set of traps and pitfalls since it is a non-trivial effort to ensure that all cryptographic and system aspects are well-addressed. A successful approach for addressing this challenge is to focus on open-source solutions because they can be scrutinized by a large community of developers. To the best of our knowledge, we are the first to offer a comprehensive study that compares the performance of these tools from the application utilization and system perspectives.

Through this work, we develop an evaluation framework to compare privacy-preserving tools to obtain a nuanced picture of the tradeoffs in data analysis where the tools are used. The framework is implemented on Docker [25], which is compatible with the dominant operating systems of Windows, Linux, and iOS and thus offers flexibility to programmers in terms of software reuse purposes. The designed framework uses the proposed evaluation criteria to quantify an analysis’ utility loss and system overhead compared to a non-private benchmark. Using the devised framework, we evaluate the most relevant open-source tools and compare their performance on DP data analysis. Through the evaluation results, we provide insights into the studied DP tools. Our study can facilitate the selection of DP tools by developers according to their needs and use cases.

For developers looking at accumulating general statistics about categorical or continuous datasets, Google DP seems promising, with a margin of error from about 0.1% to 2% for simple queries (SUM, COUNT, AVG) with ϵ⪆1.5. Given the same set of queries and ϵ values, Smartnoise provides an error of about 0.5% to 5%. We also note that Smartnoise offers better accuracy for HISTOGRAM queries, with an error below ca. 10% compared to Google DP’s about 15%, on *Health Survey*.For developers looking at building DP machine learning models, both Opacus and Tensorflow Privacy (TFP) show promising results, obtaining data utility below around 6% error for ϵ≥1.0 on *Parkinson* and *Health Survey*. Nevertheless, TFP manages to obtain around two times better data utility than Opacus given maximum data size and ϵ=3.0. Diffprivlib, on the other hand, outperforms both tools on *Health Survey* given maximum data size and ϵ=3.0. It should, however, be noted that Diffprivlib did not manage to generate any useful results on continuous data and is also limited to linear regression and logistic regression models, while TFP and Opacus offer building custom neural networks, allowing developers to build complex models for a variety of problems.

For the sake of supporting the scientific process in the area and further development, we release our work as open-source software (https://github.com/anthager/dp-evaluation, accessed on 30 May 2023). This enables the reuse of our work in further evaluation of the considered tools and beyond. We hope that this work can provide a landscape of the pros and cons of the existing open-source privacy tools and intuitive knowledge to practitioners on how to leverage them in their privacy protection service development, and ultimately narrow the gap between theoretical and applied research on DP.

### 1.5. Document Structure

We provide an overview of the evaluation settings in Section 2, where we present the proposed evaluation criteria, framework, and implementation. In that section, we explain the details of our evaluation methodology, including the specific criteria we use to assess the performance of our approach, the framework we have designed for conducting the evaluation, and the implementation details of our experimental setup. Moving forward, we present the evaluation results in Section 3, where we analyze and discuss the outcomes of our experiments. There, we provide a comprehensive presentation of the results obtained from applying our methodology. We review the related literature and compare it to our study in Section 4. Finally, in Section 5, we draw insightful conclusions and engage in discussions. We reflect on the conclusions derived from our findings and address the limitations of our proposed evaluation framework. Additionally, we outline potential future directions to further improve and expand upon our research.

## 2. Evaluation Settings

We describe our evaluation plan by explaining our criteria (Section 2.1) as well as propose our evaluation framework in Section 2.2 and present our experiment implementation in Section 2.3.

### 2.1. Evaluation Criterion

Information on individual data might be disclosed from data analyses. However, tools that mitigate this disclosure by leveraging differential privacy might ultimately cost analysis accuracy and increase the use of system resources [26,27]. Therefore, our focus is to study the difference between the DP and non-privacy-protected (NP) results that these tools produce and how this difference varies between different tools.

We adopt data utility (also referred to as accuracy) and system overhead as evaluating metrics for the considered tools. *Utility* refers to whether the data are still useful to conduct a specific functionality after the data are perturbed with DP measures, i.e.,  how much, for a given ϵ, an outcome deviates from the actual quantity it attempts to estimate, e.g.,  what degree of accuracy reduction occurs when querying on the perturbed dataset compared with the original dataset. We define the system’s *overhead* as the additional time and memory it takes to complete a DP query or train a DP-ML model versus the non-privacy-protected (NP) query or ML model. Overhead is further divided into two metrics: memory overhead and runtime overhead, which will be detailed below.

The metrics of *utility* and *overhead* illustrate what deviations can be expected from the DP results of the tools and allow for comparison amongst tools’ performances. The criterion *utility* is quantified by the deviation of the prediction error over several runs of the same experiment using the *Root Mean Square Percentage Error* (RMSPE). This is essentially the percentile difference between the DP and NP results shown in Equation (Equation 1), where *N* is the number of experiment runs, *NP* is the NP benchmark result, and *DP* is the DP result.
(1)$$\mathrm{RMPSE}=\sqrt{\frac{\sum_{n=1}^{N}{\left(\frac{NP-DP}{NP}\right)}^{2}}{N}}$$

Memory overhead is measured by comparing the worst-case memory usage between DP and NP query/ML tasks to guarantee minimum system requirements for the tools; i.e.,  the measurement shows the percentile difference between the worst-case memory usage of DP vs. NP query/ML results. Specifically, memory usage is recorded when the tools conduct DP and NP query/ML tasks. This criterion of memory overhead is considered in this evaluation since it can be noticed by users who care about the usability of the privacy tools; e.g.,  lower memory usage ultimately improves speed due to less paging, fewer cache misses, and faster structure traversals, and it also improves stability by reducing virtual and physical out-of-memory aborts. Moreover, for specific tools that use an external database, including Smartnoise and Google DP, the memory usage of the database container is also recorded. This recording will show how the tools affect the memory usage during load on the database since we are running queries on a high frequency during this process.

*Runtime overhead* is measured by comparing the time passed before and after entering the critical section, i.e.,  the part where the tool conducts an ML task or runs a query. To minimize external impact and obtain reliable and comparable results throughout the experiments, we aim to eliminate operations like initialization or saving results to the highest extent. Similar to the memory usage, we obtain the difference between the time passed before and after, comparing the DP and NP time usage. However, instead of comparing the maximum runtime, we average the results by applying RMSPE, shown in Equation (Equation 1).

### 2.2. Evaluating Framework

We construct our evaluating framework in pursuit of insights into the impact of DP tools’ privacy measures on accuracy and system resource usage. To this end, we measure the difference between DP and NP results regarding our evaluation metrics: data utility and system overhead. In addition, we vary two parameters that affect these metrics, the dataset size (Table 2) and privacy budget (Table 3), which is denoted by ϵ, to further illuminate the privacy–utility tradeoff induced by differential privacy measures.

We study the impact of ϵ and dataset size since they are the parameters that trade between privacy protection and data utility. We illuminate this tradeoff to provide developers or DP service practitioners, e.g.,  healthcare institutions and companies, with practical results to make educated choices when applying these tools. The set of sizes for considered datasets where the evaluation experiments are conducted is listed below.

Our selection of ϵ values in the evaluation takes into consideration recommendations both from research works and practical settings in the industry. While analytical research has evaluated DP algorithms using a privacy budget ranging from 0.01 to 7, practitioners prefer a narrow scale. Microsoft, in collaboration with OpenDP, explains in their product *Azure* that privacy budgets are typically set between 1 and 3 to limit the risk of re-identification (https://docs.microsoft.com/en-us/azure/machine-learning/concept-differential-privacy#differential-privacy-metrics, accessed on 30 May 2023). They state that ϵ values below 1 provide full plausible deniability and that values above 1 come with a higher risk of disclosing the actual data.

In order to cover a wide range of privacy–utility tradeoff results, we use the practice of ϵ as guidelines for selecting a set of ϵ in our evaluation experiments. The set of ϵ values considered is listed in Table 3.

Since different DP tools function and conduct computation with different techniques, we group the tools by the service that they offer into two domains, *statistical queries* and *machine learning*. This categorization allows for a reasonable comparison of tool performance within each group of tools.

For the evaluation of statistical query tools, we select a set of queries to conduct on each column in the dataset, as listed in Table 4. Note that the HISTOGRAM queries are only conducted on columns composed of categorical values since the statistics of the categorical values can be sorted into buckets. For machine learning tools, we carry out linear regression tasks since regression is the only service that all the considered ML tools have in common.

To clarify how we conduct our experiments for the different tools, we present a high-level diagram shown in Figure 1 illustrating how data flow through the different tools and generate results.

### 2.3. Experiment Implementation

We aim to enable the reuse of our framework to the full extent and allow users to develop and test our code in any environment without installing and handling dependencies locally. Therefore, we build a collective framework for all the tools, focusing on usability and portability.

To make sure that our code works across different environments, we develop the evaluation framework on Docker [25], which packages each tool package and its dependencies in a virtual container. Docker also eases the memory usage measurements by exposing a RESTful API running on the host system through a UNIX socket, from which metrics such as memory usage can be fetched.

Each tool evaluated is packaged in its own Docker image together with all the packages the tool depends on, along with the framework code. This implementation empowers evaluations using the framework to run on any Linux, Windows, or macOS computer. The complete evaluation, therefore, only requires Docker version 20.10.6 or higher.

To visualize how the framework is constructed, we present a diagram in Figure 2. It shows the dataflow in the framework as follows: (1) data and metadata (with the experiment- and hyperparameters) are loaded in Context class, which is (2) initialized in Tester script for the respective tool. (3) Tester makes use of various utils from a collective package DPEvaluation. (4) The results are saved after running Tester, whose outcomes are collected by (5) Plot Builder with Aggregator, both of which are parts of the collective utils package.

## 3. Evaluation Results

This section presents the results of our evaluation, where we show to what level prevailing functionality is affected by the use of differential privacy (DP). The results of statistical queries appear in Section 3.1 and machine learning results appear in Section 3.2. We summarize the results in Section 3.3, where we search for emerging patterns that are related to the tools’ performance.

### 3.1. Statistical Tools Assessment

This section evaluates two tools with statistical query services, namely Google Differential Privacy (Google DP) and OpenDP Smartnoise, using the two considered databases as shown in Table 2. The evaluation varies the privacy budget ϵ and data size during the experiments and looks into how the query results of SUM, AVERAGE, COUNT, and HISTOGRAM change when the DP mechanism is integrated. In the evaluation, each query runs twenty times, of which two extreme results are removed, and the remaining eighteen are averaged for analysis.

Generally, the results reflect the trend that utility increases given larger ϵ values and data sizes for both tools, yet no obverse connections among memory overhead, ϵ values, and data sizes can be observed. There also exists an opposite impact of data size on the runtime overhead for the two tools that, while larger data size results in an increase in run time for Smartnoise, it acts conversely for Google DP, although significant irregularities exist. In comparing the two query tools, we observe that Google DP offers better query accuracy than Smartnoise in all query types except HISTOGRAM and that Google DP accrues significantly less runtime than Smartnoise when comparing their DP queries against the benchmark. The overall results indicate an advantage of Google DP on Smartnoise under limited conditions. We detail the quantified evaluation results as follows.

#### 3.1.1. Data Utility

The evaluation in this part studies the differential privacy (DP) query tools’ impact on utility by comparing how DP query results differ from non-privacy-protected ones using different settings. We anticipate that a higher privacy budget, ϵ, and a larger data size will provide better utility since less noise is injected under such conditions. However, as detailed below, although the experimental results match our anticipation, we notice local irregularities, e.g.,  in the case of HISTOGRAM queries.

Figure 3 shows contour plots for each tool’s performance regarding different ϵ values and data sizes. Darker shades of blue in the plots indicate a lower RMSPE, corresponding to higher utility. Conversely, lighter shades indicate a larger RMSPE, corresponding to lower utility. Note that the plots have different RMSPE scales, implying that a shade in one plot (which indicates an RMSPE value) does not necessarily correspond to the same shade in another plot.

The contour plots demonstrate that, for simple queries of COUNT, SUM, and AVG, Google DP bears RMSPE between 0.1% and 20%, indicating 0.1–20% worse than the benchmark, while Smartnoise performs between 0.2% and 350% over the considered parameter ranges of ϵ and data size. However, Smartnoise performs better than Google DP on the HISTOGRAM queries with RMSPE between 0.5% and 60%, compared to Google DP’s between 0.2% and 250%. Such results imply Google DP’s advantage over simple query types while the reverse in the HISTOGRAM query. We also observe that HISTOGRAM queries generally have a more significant impact on data utility than other types of queries for Google DP and Smartnoise. A possible reason might be that HISTOGRAM queries expose more information about the dataset properties; as a result, more noise is injected into the HISTOGRAM results to guarantee the privacy of individuals, which in turn reduces data utility.

Moreover, the results show that HISTOGRAM queries obtain better results on *Parkinson* than on *Health Survey* for both tools, which is expected since the categorical columns in *Parkinson* have fewer bins than *Health Survey* data, thus exposing less information about *Parkinson* data properties. Consequently, less noise is injected into the results on *Parkinson* data in pursuing individuals’ privacy, and higher query accuracy is obtained.

Figure 4 shows more details on the results that higher ϵ values decrease the RMSPE (see the definition in Section 2.1), implying that higher utility of DP queries is obtained where higher ϵ values improve accuracy for all queries of SUM, AVERAGE, COUNT, and HISTOGRAM for both of our two considered datasets. This observation is quite explicit, especially for results on *Parkinson*. As anticipated, the relationship that data utility grows with the increase in data size can also be observed. Generally, the results on *Parkinson* are more consistent with our anticipation, while those for HISTOGRAM queries on *Health Survey* data show marginal levels of fluctuation.

We observed a consistent trend across Figure 3 and Figure 4, where lower epsilon values resulted in a more pronounced degradation of utility. This aligns with our expectations as smaller epsilon values correspond to tighter privacy budgets, necessitating more substantial noise addition during data analysis. Consequently, the utility of the dataset experiences a significant reduction when epsilon values are small due to the above inherent property of DP-based systems.

#### 3.1.2. Runtime Overhead

In this section, we illustrate how the runtime differs between conducting differential private (DP) queries and non-private ones by testing the tools of Google Differential Privacy and Smartnoise. Note that we vary the settings of ϵ and data size to gain evaluating results under different conditions.

For this evaluation, we anticipate that the runtime overhead might increase when DP is integrated since DP requires additional computations to conduct the query. Intuitively, the runtime for DP queries is also expected to grow with an increase in data size since more information will be processed. The results, as detailed below, demonstrate that Google DP poses less runtime than Smartnoise, while how the two tools are impacted by DP differs; i.e.,  Smartnoise experiences an increase when data size rises, while Google DP reacts in the opposite manner. Beyond that, we observe apparent fluctuations in the results and no clear relationship between ϵ and runtime.

In Figure 5, we provide contour plots for each tool’s runtime overhead regarding ϵ and data size. The plots show that large data sizes generally increase the runtime for Smartnoise, i.e.,  between 400% and 500% RMSPE for the smallest data sizes and about 450% to 550% for the largest. In contrast, the Google DP performance does not follow our anticipation, with RMPSE between 110% and 150% for the smallest data size and 100% to 130% for the largest. Overall, Google DP outperforms Smartnoise, which runs around 400% to 500% slower when conducting DP queries compared to Google DP, which runs around 100% slower. A possible reason is that Google DP performs more efficient DP calculations using a plugin inside the database compiled to native code (Section 1.2.2). Therefore, the runtime might improve since no additional layer operates between the database and the analyst that conducts the queries. In comparison, Smartnoise implements pre-processing of queries before communicating with the database (Section 1.2.1), which might negatively impact the runtime.

Figure 6 shows how ϵ impacts query accuracy for both Google DP and Smartnoise for both datasets. It reveals that higher ϵ values do not necessarily decrease or increase the RMSPE, indicating that ϵ values do not generally impact the runtime of DP queries. Although slight local fluctuations exist, this observation is quite explicit and holds for both tools and datasets, especially obvious for Google DP on *Health Survey*.

#### 3.1.3. Memory Overhead

This section investigates how DP impacts memory usage when running statistical queries on Google DP and Smartnoise. We use various ϵ and data sizes to see how the results differ under different settings. To gain a nuanced result, we measure the memory usage in both the container of the Postgres database where noise is added and processed data are stored and the container that processes the issued private or non-private queries to the database.

The memory overhead is measured by comparing the worst-case memory usage between DP and NP query or ML tasks to guarantee minimum system requirements for the tools; i.e., the measurement shows the percentile difference between the worst-case memory usage of DP vs NP query or DP-ML vs NP-ML tasks, denoted as δ. Memory usage is recorded during the time that the tools conduct DP and NP query or ML tasks. This is measured by retrieving a stream of the memory usage of the Docker containers that run the evaluation process in our evaluation. This criterion of memory overhead is considered in this evaluation since it might affect the usability of the privacy tools; e.g., lower memory usage ultimately improves speed due to less paging, fewer cache misses, and faster structure traversals, and it also improves stability by reducing virtual and physical out-of-memory aborts. Moreover, for specific tools that use an external database, including Smartnoise and Google DP, the memory usage of the database container is also recorded, which means that we will present two different memory evaluations for these tools, both for the evaluation container and the database container. This will show how the tools affect the memory usage during load on the database since we are running queries on a high frequency during this process.

In this evaluation, we anticipate the memory overhead for running queries to increase when DP is integrated, and that the memory overhead grows as the data size rises since more data are involved in the calculation procedures. As elaborated below, the results show that DP generally poses an additional memory usage of less than 3% for the operations in the Postgres database, while the processing of private queries poses 5–40% extra memory consumption. However, the results disclose no clear relationship between ϵ, data size, and memory overhead, and fluctuations of different levels exist throughout the results. In comparison, there is no apparent advantage of one tool over another regarding memory overhead, yet Google DP slightly outperforms Smartnoise on the *Health Survey* data composed of categorical variables.

Figure 7 shows contour plots for each tool’s performance on memory overhead regarding different ϵ values and data sizes. While the plots reveal no explicit patterns, the results on HISTOGRAM manifest a slight trend for the Postgres database operation in Smartnoise that memory overhead grows with an increase in data size (Figure 7d,h), which holds for both datasets. However, we also observe a significant impact of data size on the query processing compared to that of ϵ (Figure 7a,c,e,g), where the value of ϵ does not necessarily affect memory overhead, and the data size irregularly influences the memory overhead.

Figure 8 details the impact of DP on the Postgres database, from which we cannot summarize any clear correlations between ϵ, data size, and memory overhead. In general, the induced memory usage on the operation of the Postgres database is notably low (⪅3%) for both tools and datasets, which indicates a marginal impact. In comparison, although irregularities exist, Google DP shows lower peaks (less than two) in the delta axis than Smartnoise (between two and three) in the *Health Survey* experiments, implying an advantage of Google DP on categorical dataset over Smartnoise. In contrast, this advantage is not evident in the *Parkinson* evaluation.

Figure 9 describes how DP impacts the processing of queries issued to the Postgres database regarding memory overhead. We cannot observe any general relationship between ϵ, data size, and memory overhead through this figure. However, the results show an overhead of 5–40%, indicating more memory consumption of querying procedures than database operations with less than 3% memory overhead (see Figure 8). The results also suffer from fluctuations, especially for the HISTOGRAM query on *Parkinson* by Google DP, whereas Smartnoise performs more stably, although no better in that case.

### 3.2. Machine Learning Tools Assessment

We evaluate three machine learning tools with the provision of DP service, i.e.,  Tensorflow Privacy, Opacus, and Diffprivlib, and investigate how their private models differ from non-private ones learned from two datasets as shown in Table 1. This evaluation considers the regression model for all the tools, which we instantiate as a linear regression model since regression is the only functionality the considered tools hold in common. Furthermore, we vary the privacy budget ϵ and data size in the evaluation to see how the results differ regarding utility, runtime overhead, and memory overhead defined in Section 2.1. Note that each model training runs ten times, of which two extrema results are removed, and the remaining eight are averaged for analysis.

The results generally manifest the trend that the integration of DP in model training induces model accuracy reduction, and this reduction lowers with an increase in either ϵ or data size, which holds for all tools on both *Parkinson* and *Health Survey* data except Diffprivlib on *Parkinson* data, where no useful result is obtained. We also observe that Tensorflow Privacy poses less memory overhead for the larger data sizes of the considered datasets, while no clear relationship exists between ϵ, data size, and runtime overhead. In comparison, Opacus induces less model accuracy reduction than Tensorflow Privacy and Diffprivlib, given ϵ≤0.5 for both datasets, while Tensorflow Privacy outperforms Opacus and Diffprivlib on the continuous dataset of *Parkinson* within a wide range of privacy budget (0.5≤ϵ≤3.0). The results also indicate that Tensorflow Privacy poses less runtime, and Opacus adds less memory usage when DP is integrated into model training. The quantitative evaluating results are detailed in the following.

#### 3.2.1. Data Utility

Evaluation in this section investigates how a differentially private machine-learning model differs from a benchmark regarding model accuracy to show the tradeoff between privacy and utility when a machine learning task is combined with differential privacy (DP) under different experimental settings.

In this evaluation, we expect that higher ϵ values and larger data sizes will provide better utility for the considered datasets since such conditions cause less noise added during DP model training. As anticipated, the results show that the trained machine learning model brings better accuracy as ϵ and data size increase. However, along with this expected trend, there also exists local irregularities, and Diffprivlib incurs severe accuracy reduction on the Parkinson data making it far from useful in that case. We further detail the evaluation results below.

The contour plots in Figure 10 describe how the learned model’s utility measured by RMSPE varies regarding ϵ and data size. In general, the plots corroborate that utility grows with larger ϵ and data size, though there are irregularities in the *Parkinson* results by Opacus, where the data size 4000 presents a slightly increased RMSPE than the lower data size. Also, ϵ does not necessarily affect Opacus’ modeling utility on *Parkinson* when ϵ≥1.0. We also observe that in the *Health Survey* experiments, Opacus exhibits marginal less RMSPE (generally ≤6) than Tensorflow Privacy (generally ≤10), both of which perform much better than Diffprivlib (≥10 under most of the settings); However, Tensorflow Privacy provides obvious less RMSPE than Opacus and Diffprivlib on *Parkinson* data, as shown in Figure 10d–f, indicating Tensorflow Privacy’s advantage on continuous data.

Figure 11 depicts model accuracy reduction under different ϵ values, where we observe that higher ϵ values decrease the RMSPE, implying less accuracy reduction of the DP-integrated-model compared with the benchmark. This trend holds for the two considered datasets by Tensorflow Privacy and Opacus, while Diffprivlib provides RMSPE of over $10^{8}$ on the *Parkinson* data (Figure 11g–i), making it unacceptable under this scenario. Therefore, we neglect the experimental results of Diffprivlib on *Parkinson* in the following analyses. Even though, it is worth noting that Diffprivlib generates comparably equal accuracy to Opacus and Tensorflow Privacy when no DP is applied on both datasets. In comparison, Opacus outperforms Tensorflow Privacy and Diffprivlib when ϵ≤0.5. However, since RMSPE rises abruptly when ϵ decreases away from 0.5, the advantage of Opacus here becomes insignificant. In contrast, Tensorflow Privacy produces less accuracy reduction for *Parkinson* data within a wide range of ϵ (0.5–3.0), implying its better performance on continuous dataset than Opacus and Diffprivlib.

#### 3.2.2. Runtime Overhead

Evaluation of this part presents how the considered machine learning (ML) tools perform when combined with differential privacy (DP) regarding the runtime overhead induced due to DP. We vary the experimental settings to see how the results differ under various conditions.

The evaluation is anticipated to observe an increase in DP machine learning’s runtime compared with the benchmark, and we also expect that the runtime overhead grows as the data size rises in the experiments since more data are processed. The results, as detailed below, show that Tensorflow Privacy poses the least runtime increase compared with Opacus and Diffprivlib, with the general RMSPE of ≤40 in runtime for Tensorflow Privacy versus 200–230 for Opacus and 400–2000 for Diffprivlib. Note that we did not display the results of Diffprivlib on *Parkinson* data since there is no useful result generated, as elaborated in Section 3.2.1. Beyond that, we observe no clear trend between data size, ϵ, and runtime overhead.

Contour plots in Figure 12 describe each tool’s runtime overhead regarding different ϵ values and data sizes. Through the plots, we observe that both ϵ and data size affect the runtime overhead in an irregular manner, where no explicit patterns can be concluded. The results also present fluctuations and irregularities that are hard to explain, e.g.,  the abrupt increased RMSPE for Tensorflow Privacy on *Health Survey* when ϵ=2 and $data\;size=6\times 10^{3}$, and the sudden decreased RMSPE for Opacus on *Parkinson* when $data\;size=3\times 10^{3}$. Even though, it is clear that Tensorflow Privacy incurs less RMSPE due to DP in machine learning, compared with Opacus and Diffprivlib on both datasets.

Figure 13 illustrates the induced runtime under different ϵ values on the *Health Survey* and *Parkinson* data for all the ML tools. The graphs reveal no relationship between ϵ and runtime during model training of all the considered ML tools, while the results of the *Health Survey* demonstrate a stable induced runtime by Tensorflow Privacy and Opacus. However, as shown in Figure 13a, local irregularity exists for Tensorflow Privacy. Overall, Tensorflow Privacy significantly outperforms Opacus and Diffprivlib regarding runtime on both the considered datasets.

#### 3.2.3. Memory Overhead

This section investigates the additional memory usage posed due to the integration of differential privacy (DP) in machine learning (ML). We conduct experiments using various settings to look into how the considered ML tools perform in DP ML models compared with non-private ones.

For the experiments, we anticipate an increase in memory usage in the training of DP ML models compared with non-private ones since more DP models involve more computation during the model training. As detailed below, the experimental results demonstrate extra memory usage of different levels for all the considered ML tools due to DP. Particularly, Tensorflow Privacy suffers the most memory usage (15–20% for the *Health Survey* and above 70% for *Parkinson*) compared with Opacus (below 3.75% for the *Health Survey* and below 4.0% for *Parkinson*). On the other hand, Diffprivlib generally contributes an additional memory usage of 5–10% for the *Health Survey*, yet it does not provide useful results for *Parkinson*. Overall, Opacus shows more advantage in memory overhead for DP machine learning.

Figure 14 shows contour plots of each tool’s memory overhead regarding different ϵ values and data size. From the plots, we can observe that TFP has a memory overhead of about 90% for the *Parkinson* dataset for nearly all the data size and ϵ, and about 20% for the *Health Survey* dataset (Figure 14a,e), although there is an unexpected decreased memory usage for the highest data size of both *Health Survey* and *Parkinson*. Opacus (Figure 14b,d), on the other hand, has less than 5% memory overhead in both of the datasets, while irregularities exist for some data size and ϵ combinations for the *Health Survey*, and an increased memory usage around the data size of 2000 for *Parkinson*. Diffprivlib gains memory overhead mainly around 10% and experiences a slightly decreased memory usage under the smallest data size and a significant decrease under the highest data size for the *Health Survey*. We also note that the value of ϵ has no apparent influence on the results for all three considered tools.

Figure 15 shows representative results on how memory overhead varies over different combinations of data size and ϵ. Beyond the conclusion from Figure 14, we can also observe a higher memory overhead for the *Parkinson* than that of the *Health Survey*. In contrast, Opacus performs stably for both the two datasets with the lowest memory overhead.

### 3.3. Summary of Results

We observed that the experimental results depend on the studied tasks, the selected DP tools, and their configurations. We note that some patterns emerge from the results; see Table 5. In order to facilitate the tool selection, we define the following performance criteria.

Any DP tool is inherently associated with some basic tradeoffs. For example, when processing a large dataset, the system costs, e.g., processing time and memory usage, increase. Another well-known tradeoff exists between the privacy budget and the utility of the privacy-protected data. Naturally, the evaluated tools might include additional dependencies and tradeoffs. In order to highlight such basic boundaries, Table 6 provides the values of the utility and system costs under *Favorable Experiment Settings* (FES) and *Restrictive Experiment Settings* (RES) with regard to the size of the dataset and privacy budget. Specifically for FES, the size of the dataset is 9358 records and ϵ=3.0 for the categorical data. The size is 5499 records and ϵ=3.0 for the continuous data. For the case of RES, the size is 1000 records and ϵ=0.1 both for categorical and continuous data. For a more *Detailed version of Performance Comparison* (DPC), we provide the utility and system costs that the tools have when considering an exhaustive set of pairs of dataset size (Table 2) and privacy budget (Table 3).

With the defined criteria and the performance measurement under each criterion, we facilitate a need-based selection of DP tools, e.g., the highest utility, the lightest computing resource requirement, and the least running time. Table 6 suggests that TensorFlow Privacy and PyTorch Opacus perform equivalently well for machine learning tasks. Also, Google DP works well for statistical queries.

## 4. Related Work

In this section, we conduct a review of the existing literature concerning the performance evaluation of privacy tools that employ differential privacy (DP). Despite the abundance of studies on DP, to the best of our knowledge, none of them have provided a comprehensive comparative study of the available open-source tools applicable in practical scenarios. Furthermore, they often lack sufficient guidance on the application and configuration of DP tools for privacy preservation.

### 4.1. Statistical Queries

As mentioned, the set of statistical queries, in this work, refer to the operation of SUM, COUNT, AVERAGE, and HISTOGRAM. One DP query engine that has been integrated into the industry is Flex [28]. The open-source library for Flex is deprecated (https://github.com/uber-archive/sql-differential-privacy, accessed on 30 May 2023). However, their evaluation remains relevant to provide clues on evaluating privacy tools.

Johnson et al. [28] evaluated Flex using an SQL-compatible interface, which makes it convenient to put the interface in front of any already deployed SQL-compatible database and, in turn, lowers the bar of adoption. Their evaluation uses a large set of real-world queries run by Uber’s data engineers in production, which provides insights into how Flex would perform in the industry. However, this evaluation merely benchmarks one ϵ value (ϵ=0.1) and presents only one single value of the additional overhead of 4.86 ms corresponding to 0.03 % of the average execution time of their non-privacy-protected queries. Furthermore, the 4.86 ms overhead does not include the pre-collection of *frequent join* attributes, which has to be updated each time the underlying data are updated. Such an evaluating procedure induces the risk that the actual overhead might be more significant than presented by Johnson et al. Moreover, it is not clear whether overhead changes when settings differ, e.g., in dataset sizes, privacy configurations, or queries, etc.

The DP tools of Smartnoise [27] and Google DP [21] are integrated into private statistical-query services. Smartnoise results from years of cumulative experience building and deploying privacy tools for research and has recently become a tool in Microsoft’s privacy ecosystem. However, even though the open-source software community provides transparency and demonstrates examples, no comparative research has been conducted on the tool’s query engine.

Google DP has been evaluated alongside Flex [28] and PinQ, which is a research project on privacy-integrated queries (PinQ). PinQ provides a programming language and execution platform in which all expressible programs satisfy DP [8,21]. The evaluation of Google DP included 1,000,000 runs with various aggregate functions, with a fixed ϵ value of 0.1, where a benchmark TPC-H dataset was used. Meanwhile, in the comparison, Flex and PinQ were run only 10,000 times due to performance concerns. This evaluation points out that, compared with Google DP, Flex and PinQ cannot enforce contribution bounds for databases where one single user can contribute multiple samples, leading to query results that are not DP. Furthermore, because Flex or PinQ assume that the underlying database is associated with at most one record per user, their performance comparison with Google DP that supports the contribution of multiple samples can be problematic. The evaluation of Google DP merely considered a single value of privacy budget. Our extensive evaluation uses broad evaluation settings, i.e., different privacy budgets and dataset sizes.

Garrido et al. [29] compared the performances of Google DP, SmartNoise, diffprivlib, diffpriv, and Chorus with respect to the features offered to the users. However, Garrido et al. do not consider issues related to machine learning as we do. Hay et al. [30] proposed DPBench, which is an evaluation framework for privacy-preseving algorithms but only in the context of histograms.

Our study encompassed a broad range of query functions, including SUM, COUNT, AVERAGE, and HISTOGRAM. Previous research primarily concentrated on smaller or different sets of functions. For instance, Garrido et al. [29] examined COUNT, SUM, AVERAGE, and VAR, but not on HISTOGRAM as we do. Hay et al. [30] and Bu et al. [31] focused specifically on COUNT, while Xu et al. [32] emphasized HISTOGRAM.

### 4.2. Machine Learning

DP machine learning (ML) has gained attention at companies like Google, Facebook, and IBM. Investigations have been conducted on the performance of DP stochastic gradient descent (SGD) [16], which is a dominant algorithm for private training of ML models. Nevertheless, DP-SGD can increase the training time significantly compared to non-private SGD [31]. In a recent study, Subramani et al. [15] reduced the runtime overhead when executing DP-SGD. They implemented the functionality in the open-source library of Tensorflow Privacy by exploiting language primitives [31]. Microsoft has shown support in the DP-ML field by implementing Opacus and demonstrating the impact of epsilon and dataset size on DP-ML [26]. To the best of our knowledge, we are the first to extensively compare these DP ML tools.

Beyond improvements regarding performance, there is also work to improve the privacy guarantees of DP-ML [33] and evaluate different means of DP with different privacy budgets, which shows how the tradeoff varies between utility and privacy protection under different settings. Their study focuses on gradient perturbation mechanisms, e.g., DP-SGD, and uses Tensorflow Privacy to evaluate Rényi DP among others [34]. It demonstrates that the privacy guarantees of DP for ML implementations may provide unacceptable balances of the tradeoff between utility and privacy protection. They aim to find epsilon values that balance the tradeoff for different DP approaches rather than for different DP tools. In another study, Tramér et al. [35] provide an approach to improving performances of DP models using primarily Tensorflow and parts of the Opacus library. They point out that prior works have underestimated guarantees of utility and privacy protection as well as demonstrated that solid privacy may come at only a nominal cost of inaccuracy by tailoring the training to the data. However, their results, to the best of our knowledge, have not yet been generalized as a tool that can be easily leveraged by the developer community. Our study of the tradeoff between utility and privacy protection considers different DP-ML tools as well as facilitates the selection of DP-ML tools.

### 4.3. Bibliographical Note

A technical report version of this article can be found at [36].

## 5. Conclusions

We propose an evaluation framework for differential privacy (DP) tools and offer evaluation for state-of-the-art open-source DP tools. We define criteria to quantify how different DP tools perform so that they can be selected. Specifically, we evaluate and measure the impact of DP on different functionalities that the studied tools provide. We use two data sources of different types to obtain a nuanced picture of how well the studied tools perform when DP is applied. The evaluation results demonstrate the degree to which the use of DP tools impacts data utility and system overhead. Our results can support practitioners who consider using these tools.

### 5.1. Limitations

The studied approach did not include the conduction of a white box evaluation of the DP tools since we wished to emphasize the most typical use case scenarios. Furthermore, in our current work, we have not explicitly considered the correctness aspect of the DP tools. While our evaluation focused on performance and utility, it is essential to assess the correctness of the DP mechanisms to ensure that the privacy guarantees are effectively preserved. Such assessment requires other analytical tools and cannot be achieved using only empirical results.

### 5.2. Future Work

We invite researchers and practitioners to leverage the proposed open-source evaluation framework to explore additional aspects of DP tools, datasets, and machine-learning methods. Since the framework is publicly available in an open-source format, it can serve as a foundation for conducting comparative evaluations of emerging DP tools and analyzing their performance in different scenarios. By extending the evaluation to include a broader range of prospective tools, datasets with varying characteristics, and diverse machine-learning techniques, the readers are offered to gain a more comprehensive understanding of how future DP tools impact different datasets and applications most relevant to their individual interests.

## Figures and Tables

**Figure 1 sensors-23-06509-f001:**
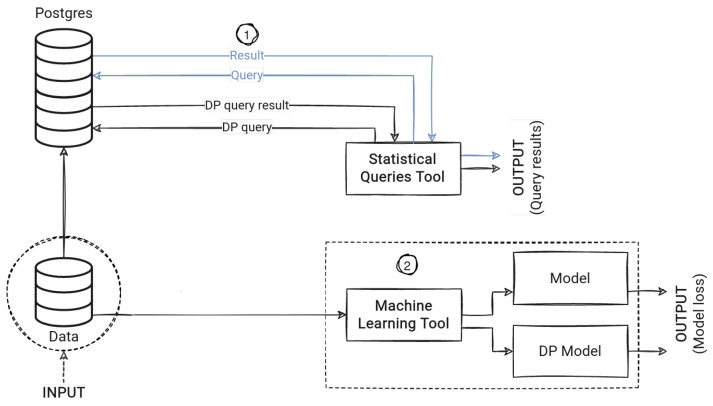
A high-level overview of the experiment flow, showing (1) DP queries and non-privacy-protected (NP) queries being conducted on Postgres and (2) ML tasks conducted for DP models and non-privacy-protected models.

**Figure 2 sensors-23-06509-f002:**
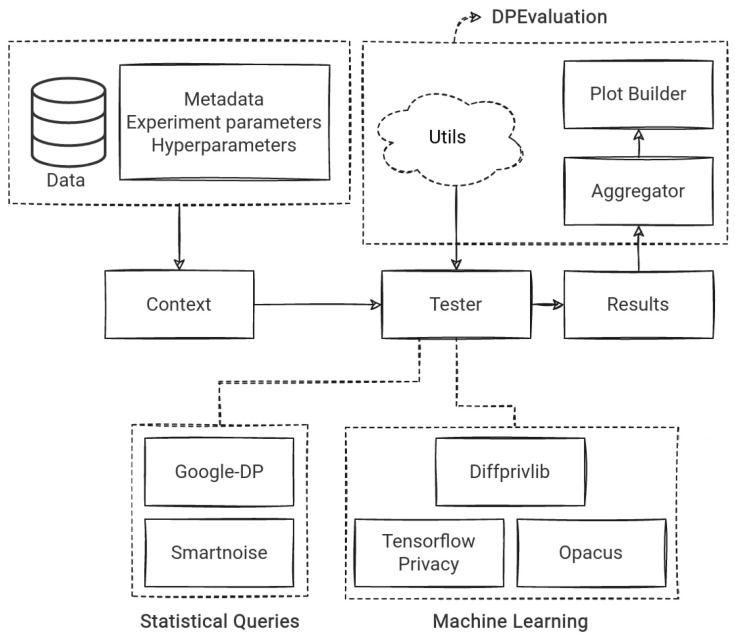
High-level overview of experiment implementation, showing how data flow in the evaluation framework.

**Figure 3 sensors-23-06509-f003:**
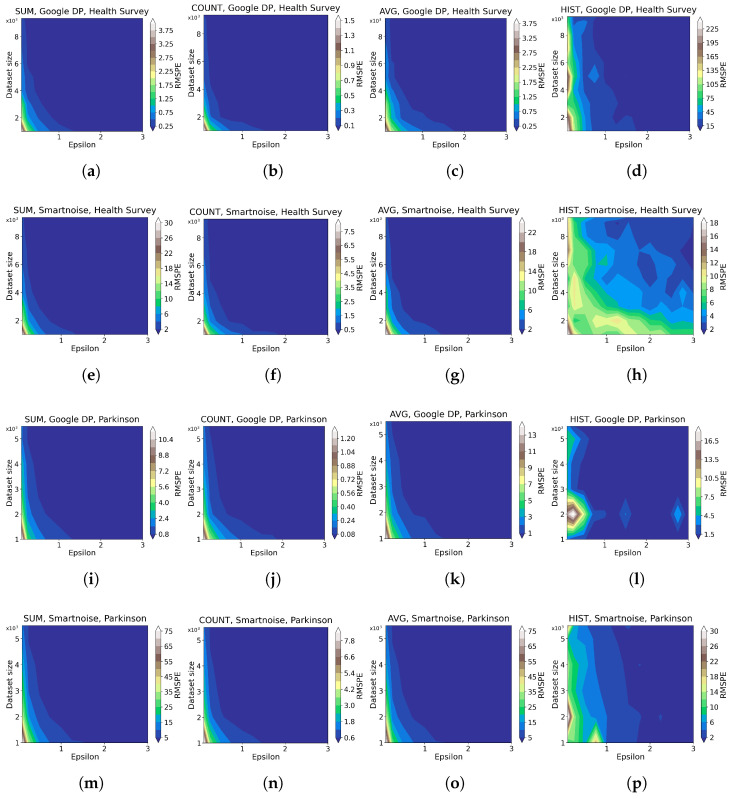
Contour plots for the evaluation of statistical query tools on utility when DP is integrated for different data sizes (Table 2), ϵ values (Table 3), and queries (Table 4). RMSPE is defined in Section 2.1. The sub-plots (**a**–**d**) present the results by Google DP on *Health Survey* for the query SUM, COUNT, AVERAGE, and HISTOGRAM, respectively. The sub-plots (**e**–**h**) present the results by Smartnoise on *Health Survey* for the query SUM, COUNT, AVERAGE, and HISTOGRAM, respectively. The sub-plots (**i**–**l**) present the results by Google DP on *Parkinson* for the query SUM, COUNT, AVERAGE, and HISTOGRAM, respectively. The sub-plots (**m**–**p**) present the results by Smartnoise on *Parkinson* for the query SUM, COUNT, AVERAGE, and HISTOGRAM, respectively.

**Figure 4 sensors-23-06509-f004:**
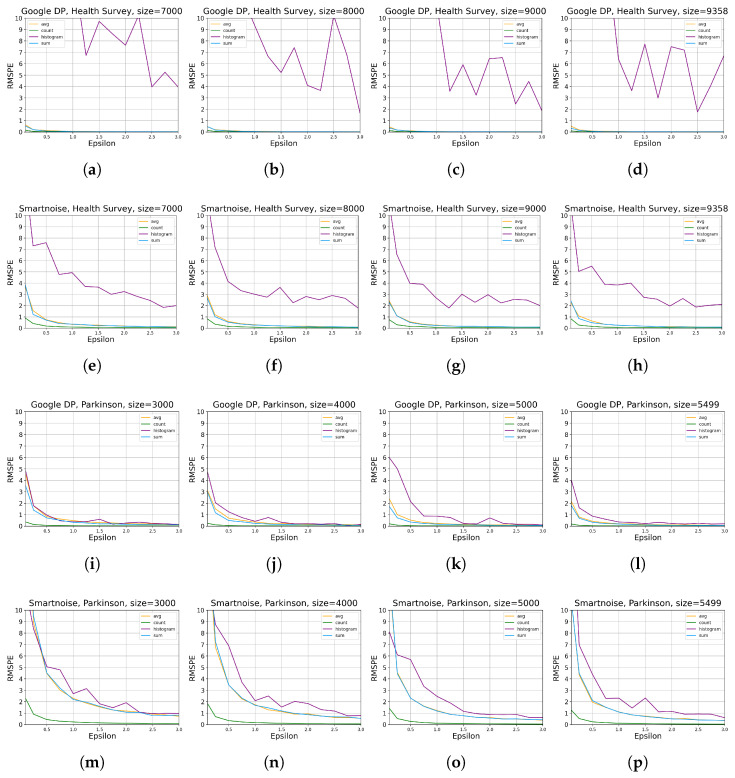
The evaluation results of statistical query tools on utility for different data sizes (Table 2), ϵ values (Table 3), and queries (Table 4). RMSPE is defined in Section 2.1. The sub-plots (**a**–**d**) present the results by Google DP on *Health Survey* with data size of 7000, 8000, 9000, and 9358, respectively. sub-plots (**e**–**h**) present the results by Smartnoise on *Health Survey* with data size of 7000, 8000, 9000, and 9358, respectively. The sub-plots (**i**–**l**) present the results by Google DP on *Parkinson* with data size of 3000, 4000, 5000, and 5499, respectively. The sub-plots (**m**–**p**) present the results by Smartnoise on *Parkinson* with data size of 3000, 4000, 5000, and 5499, respectively.

**Figure 5 sensors-23-06509-f005:**
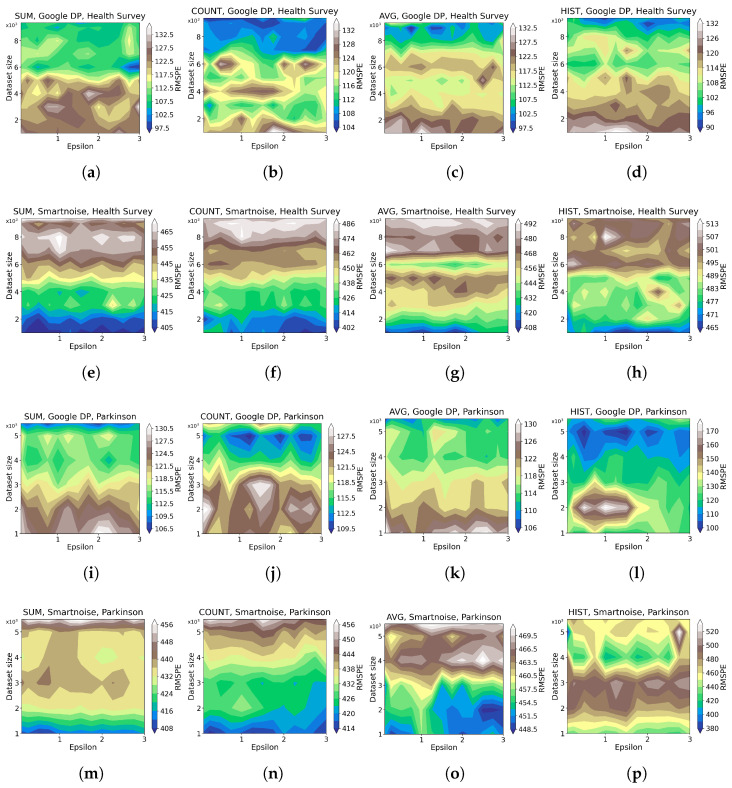
Contour plots for the evaluation of statistical query tools on runtime overhead for different data sizes (Table 2), ϵ values (Table 3), and queries (Table 4). RMSPE is defined in Section 2.1. The sub-plots (**a**–**d**) present the results by Google DP on *Health Survey* for the query SUM, COUNT, AVERAGE, and HISTOGRAM, respectively. sub-plots (**e**–**h**) present the results by Smartnoise on *Health Survey* for the query SUM, COUNT, AVERAGE, and HISTOGRAM, respectively. The sub-plots (**i**–**l**) present the results by Google DP on *Parkinson* for the query SUM, COUNT, AVERAGE, and HISTOGRAM, respectively. The sub-plots (**m**–**p**) present the results by Smartnoise on *Parkinson* for the query SUM, COUNT, AVERAGE, and HISTOGRAM, respectively.

**Figure 6 sensors-23-06509-f006:**
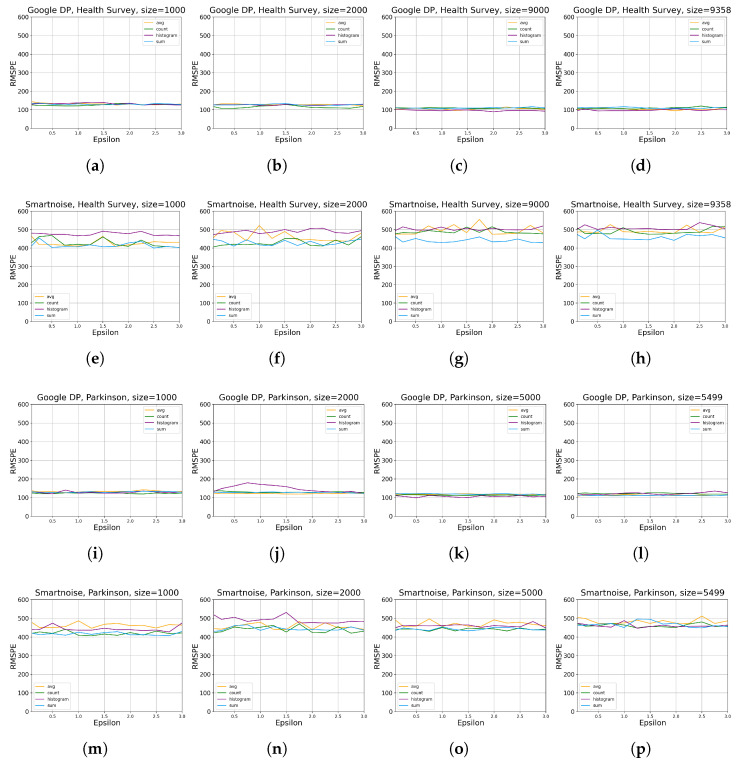
The evaluation results of statistical query tools on runtime overhead when DP is integrated for different data sizes (Table 2), ϵ values (Table 3), and queries (Table 4). RMSPE is defined in Section 2.1. The sub-plots (**a**–**d**) present the results by Google DP on *Health Survey* with data size of 7000, 8000, 9000, and 9358, respectively. sub-plots (**e**–**h**) present the results by Smartnoise on *Health Survey* with data size of 7000, 8000, 9000, and 9358, respectively. The sub-plots (**i**–**l**) present the results by Google DP on *Parkinson* with data size of 3000, 4000, 5000, and 5499, respectively. The sub-plots (**m**–**p**) present the results by Smartnoise on *Parkinson* with data size of 3000, 4000, 5000, and 5499, respectively.

**Figure 7 sensors-23-06509-f007:**
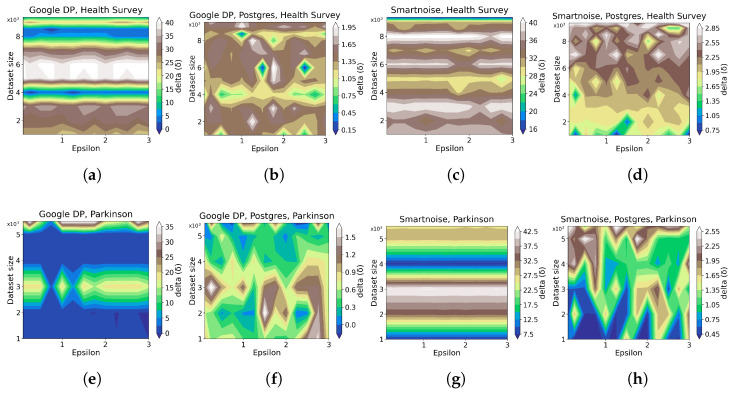
Contour plots for the evaluation of statistical query tools on memory overhead for different data sizes (Table 2), ϵ values (Table 3), and queries (Table 4). δ is defined in Section 2.1. The sub-plots (**a**–**d**) present the results on *Health Survey* by Google DP, Google DP with Postgre, Smartnoise, and Smartnoise with Postgres, respectively. sub-plots (**e**–**h**) present the results on *Parkinson* by Google DP, Google DP with Postgre, Smartnoise, and Smartnoise with Postgres, respectively.

**Figure 8 sensors-23-06509-f008:**
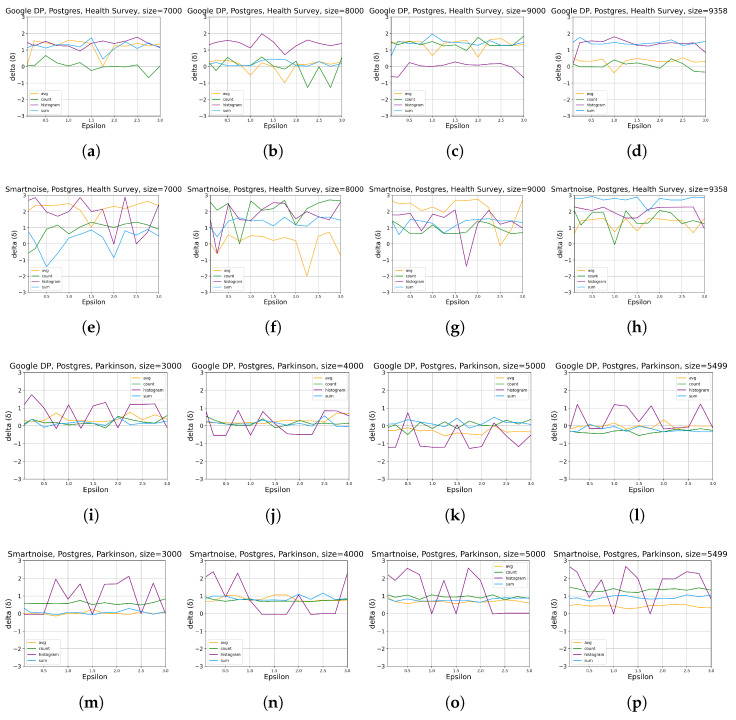
The evaluation results of statistical query tools on memory overhead when DP is integrated for different data sizes (Table 2), ϵ values (Table 3), and queries (Table 4), specifically the results on memory overhead in the database (Postgres) that the queries are conducted on. δ is defined in Section 2.1. The sub-plots (**a**–**d**) present the results by Google DP on *Health Survey* with data size of 7000, 8000, 9000, and 9358, respectively. sub-plots (**e**–**h**) present the results by Smartnoise on *Health Survey* with data size of 7000, 8000, 9000, and 9358, respectively. The sub-plots (**i**–**l**) present the results by Google DP on *Parkinson* with data size of 3000, 4000, 5000, and 5499, respectively. The sub-plots (**m**–**p**) present the results by Smartnoise on *Parkinson* with data size of 3000, 4000, 5000, and 5499, respectively.

**Figure 9 sensors-23-06509-f009:**
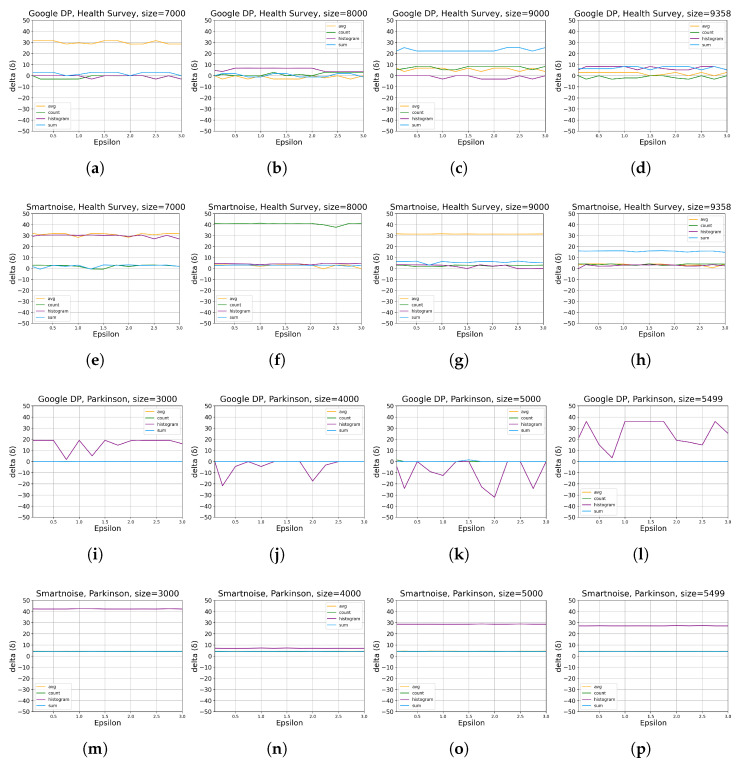
The evaluation results of statistical query tools on memory overhead when DP is integrated for different data sizes (Table 2), ϵ values (Table 3), and queries (Table 4). δ is defined in Section 2.1. The sub-plots (**a**–**d**) present the results by Google DP on *Health Survey* with data size of 7000, 8000, 9000, and 9358, respectively. sub-plots (**e**–**h**) present the results by Smartnoise on *Health Survey* with data size of 7000, 8000, 9000, and 9358, respectively. The sub-plots (**i**–**l**) present the results by Google DP on *Parkinson* with data size of 3000, 4000, 5000, and 5499, respectively. The sub-plots (**m**–**p**) present the results by Smartnoise on *Parkinson* with data size of 3000, 4000, 5000, and 5499, respectively.

**Figure 10 sensors-23-06509-f010:**
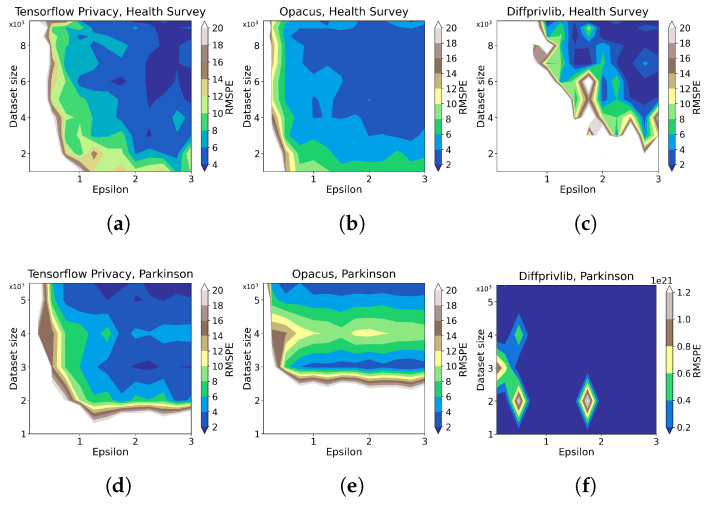
Contour plots for the evaluation of DP ML tools on data utility for different data sizes (Table 2) and ϵ values (Table 3). RMSPE is defined in Section 2.1. The sub-plots (**a**–**c**) present the results on *Health Survey* data under Tensorflow, Opacus, and Diffprivlib, respectively, and (**d**–**f**) present the results on *Parkinson* data under Tensorflow, Opacus, and Diffprivlib, respectively.

**Figure 11 sensors-23-06509-f011:**
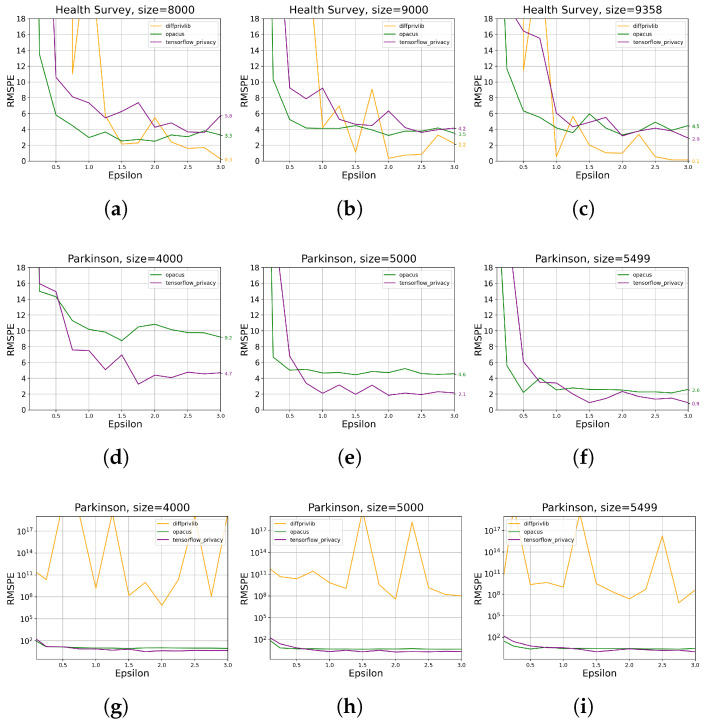
The evaluation results of DP ML tools on data utility for different data sizes (Table 2) and ϵ values (Table 3). RMSPE is defined in Section 2.1. The sub-plots (**a**–**c**) present the results on *Health Survey* with data size 8000, 9000, and 9358, respectively. The sub-plots (**d**–**f**) present the results on *Parkinson* with data size 4000, 5000, and 5499, respectively. The sub-plots (**g**–**i**) repeat the results in (**d**), (**e**), and (**f**), respectively, with a larger y-axis scale.

**Figure 12 sensors-23-06509-f012:**
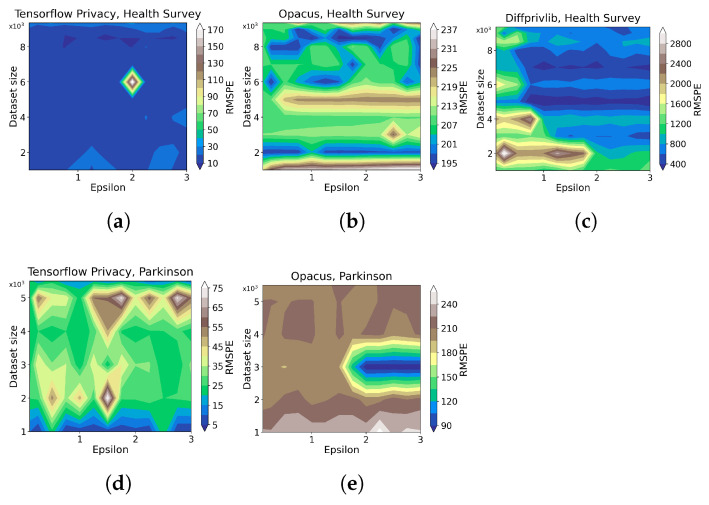
Contour plots for the evaluation of DP ML tools on runtime overhead for different data sizes (Table 2) and ϵ values (Table 3). RMSPE is defined in Section 2.1. The sub-plots (**a**), (**b**), and (**c**) present the results on *Health Survey* data under Tensorflow, Opacus, and Diffprivlib, respectively, and (**d**) and (**e**) present the results on *Parkinson* data under Tensorflow and Opacus, respectively.

**Figure 13 sensors-23-06509-f013:**
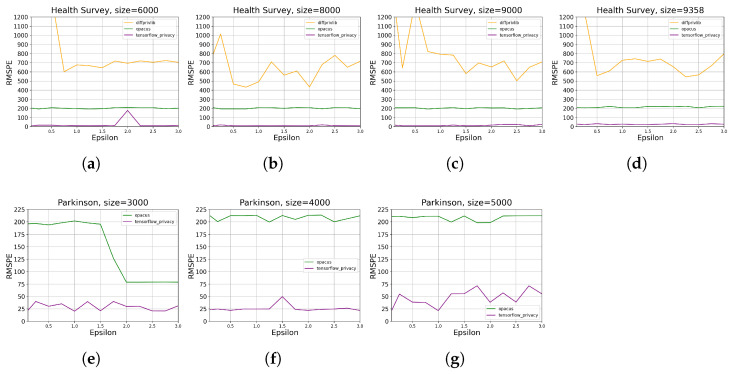
The evaluation results of DP ML tools on runtime overhead for different data sizes (Table 2) and ϵ values (Table 3). RMSPE is defined in Section 2.1. The sub-plots (**a**–**d**) present the results on *Health Survey* with data size of 6000, 8000, 9000, and 9358, respectively. The sub-plots (**e**), (**f**), and (**g**) present the results on *Parkinson* with data size of 3000, 4000, and 5000, respectively.

**Figure 14 sensors-23-06509-f014:**
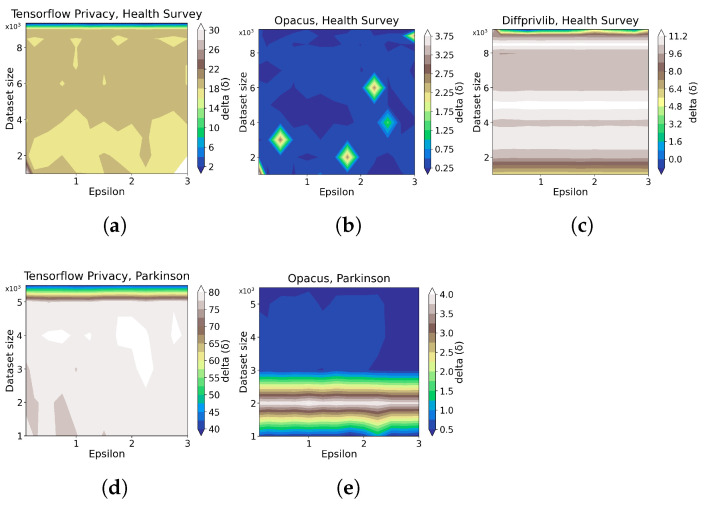
The results of the impact of ML tools on memory overhead for different data sizes (Table 2) and ϵ values (Table 3). RMSPE is defined in Section 2.1. The sub-plots (**a**), (**b**), and (**c**) present the results on *Health Survey* data under Tensorflow, Opacus, and Diffprivlib, respectively. The sub-plots (**d**) and (**e**) present the results on *Parkinson* data under Tensorflow and Opacus, respectively.

**Figure 15 sensors-23-06509-f015:**
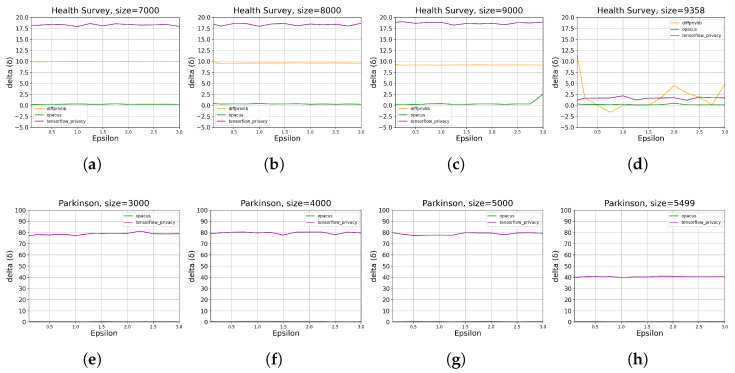
The results of the impact of ML tools on memory overhead for different data sizes (Table 2) and ϵ values (Table 3). The sub-plots (**a**–**d**) present the results on *Health Survey* with data size of 6000, 8000, 9000, and 9358, respectively. sub-plots (**e**–**h**) present the results on *Parkinson* with data size of 3000, 4000, 5000, and 5499, respectively.

**Table 1 sensors-23-06509-t001:** Evaluation strategies in this work.

		Machine Learning Domain	Statistical Query Domain
Environment		• Docker	
Task to evaluate		• Regression	Sum Count Average Histogram
Influencing factors		Dataset size Privacy budget, ϵ	
Evaluation criterion	Utility	Prediction accuracy reduction compared with non-private machine learning.	Decreased query accuracy due to privacy measures.
	Overhead	Extra resource consumption induced by differential privacy, including Execution time Memory consumption	
Database to evaluate on		US health reform monitoring survey data UCI Parkinson dataset	

**Table 2 sensors-23-06509-t002:** List of dataset sizes for each dataset.

Dataset Sizes
Health Survey size ∈ {1000, 2000, 3000, 4000, 5000, 6000, 7000, 8000, 9000, 9358}
Parkinson size ∈ {1000, 2000, 3000, 4000, 5000, 5499}

**Table 3 sensors-23-06509-t003:** The privacy budget, ϵ, values.

ϵ Values
ϵ∈ {0.1, 0.25, 0.5, 0.75, 1.0, 1.25, 1.5, 1.75, 2.0, 2.25, 2.5, 2.75, 3.0}

**Table 4 sensors-23-06509-t004:** List of queries.

QUERIES
Queries qs∈ {SUM, AVG, COUNT, HISTOGRAM}

**Table 5 sensors-23-06509-t005:** Summary of performance patterns emerging from the evaluation results of the studied tools for machine learning (ML) and statistical query (SQ).

Tool Name		Patterns Revealed
ML	Diffprivlib	• utility and runtime overhead grow with ϵ and data size on categorical data
	PyTorch Opacus	utility grows with ϵ on categorical and continuous data utility grows with data size on categorical data
	TensorFlow Privacy	utility grows with ϵ and data size on categorical and continuous data memory overhead grows with data size on categorical and continuous data
SQ	Google Differential Privacy	utility grows with ϵ and data size on categorical and continuous data higher runtime overhead for lower data size on categorical and continuous data
	OpenDP SmartNoise	utility grows with ϵ and data size on categorical and continuous data runtime overhead grows with data size on categorical and continuous data

**Table 6 sensors-23-06509-t006:** Tool performance comparison for the tasks of machine learning and statistical query. For each criterion, we provide the ranking (in the upper part of the table cells) of different tools associated with the numerical performance (in the lower part of the table cells) under the considered criterion.

		Performance Criteria (RMPSE)																	
		Categorical Data									Continuous Data								
		Utility FES	Runtime FES	Memory FES	Utility RES	Runtime RES	Memory RES	Utility DFC	Runtime DFC	Memory DFC	Utility FES	Runtime FES	Memory FES	Utility RES	Runtime RES	Memory RES	Utility DFC	Runtime DFC	Memory DFC
machine learning	Diffprivlib	1st 0.15	3rd 797.86	3rd 6.35	3rd 6.4 $\times \;10^{8}$	3rd 1176.20	3rd 46.21	3rd 2.76 to 1.21 $\times \;10^{8}$	3rd 603.17 to 1.15 $\times \;10^{3}$	3rd 9.11 to 16.64	Results far away from usable								
											4.6 $\times \;10^{8}$	2323.46	35.64	1.96 $\times \;10^{11}$	856.53	31.54	1.53 $\times \;10^{9}$ to 1.89 $\times \;10^{18}$	603.17 to 1.15 $\times \;10^{3}$	40.88 to 45.38
	PyTorch Opacus	3rd 4.47	2nd 221.13	2nd 0.72	1st 1283.47	2nd 217.85	1st 2.35	1st 3.44 to 7.10	2nd 197.67 to 211.96	1st 0.11 to 0.16	2nd 2.59	2nd 213.23	1st 1.77	1st 5122.05	2nd 224.95	1st 1.81	2nd 3.99 to 38.21	2nd 199.77 to 213.90	1st 0.55 to 1.68
	TensorFlow Privacy	2nd 2.91	1st 22.52	1st 6.1 $\times \;10^{-4}$	2nd 6960.15	1st 17.25	2rd 9.74	2rd 4.65 to 12.73	1st 10.30 to 14.23	2nd 0.64 to 7.06	1st 0.90	1st 35.69	2nd 15.77	2nd 6177.35	1st 1.57	2nd 28.32	1st 2.32 to 26.37	1st 21.01 to 35.25	2nd 13.48 to 26.88
statistical query	Google Differential Privacy	1st 0.61	1st 115.63	1st 8.12	1st 74.3	1st 137.37	1st 25.19	1st 0.016 to 0.31	1st 108.25 to 129.35	1st 0.05 to 1.88	1st 0.5	1st 120.32	1st 4.6 $\times \;10^{-4}$	1st 75.9	1st 126.63	1st 7.6 $\times \;10^{-5}$	1st 0.02 to 0.36	1st 115.11 to 137.43	1st 0 to 2.16
	OpenDP SmartNoise	2rd 2.9	2rd 481.11	2rd 14.08	2nd 801	2rd 413.96	2rd 29.27	2rd 0.11 to 2.08	2rd 408.74 to 466.23	2rd 0.50 to 2.75	2rd 3.9	2rd 473.63	2rd 4.56	2nd 746	2rd 558.16	2rd 4.67	2rd 0.19 to 2.24	2rd 427.19 to 465.05	2rd 5.06 to 6.47

## Data Availability

As mentioned, for the sake of supporting the scientific process in the area and further development, we release our work as open-source software via https://github.com/anthager/dp-evaluation, accessed on 30 May 2023. This enables the reuse of our work in further evaluation of the considered tools and beyond. We hope that this work can provide a landscape of the pros and cons of the existing open-source privacy tools and intuitive knowledge to practitioners on how to leverage them in their privacy protection service development, and ultimately narrow the gap between theoretical and applied research on DP.

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
