# Peer review of "Evaluation of Open-Source Tools for Differential Privacy"

_sensors, 2023, doi:10.3390/s23146509_

Round 1

Reviewer 1 Report

This article explores how to improve the trade-off between the degree of privacy protection and the ability to use protected data.  Differential privacy (DP) defines privacy protection by promising quantified indistinguishability between individuals that consent to share their privacy-sensitive information and the ones that do not. This work proposes an open-source evaluation framework for privacy protection solutions and offers evaluation for OpenDP Smartnoise, Google DP, PyTorch Opacus, Tensorflow Privacy, and Diffprivlib. In addition to studying their ability to balance the above trade-off, the authors consider discrete and continuous attributes by quantifying their performance under different data sizes. Defined criteria to quantify how different DP tools perform so that they can be selected. Evaluated and measure the impact of DP on different functionality that the studied tools provide. Two data sources of different types were used to get a nuanced picture of how well the studied tools work when using DP. The evaluation results demonstrate the degree to which the use of DP tools impacts data utility and system overhead.

The choice of the proposed methods is justified. This approach is relevant. The research is of practical importance.

The conclusions correspond to the presented evidence and arguments, and they correspond to the basic question. The references appropriately. The tables, pictures and formulas meet the requirements.

Author Response

Dear editors and reviewers,

We have carefully reviewed the comments and have revised the manuscript accordingly. Our responses are given in a point-by-point manner below. Changes to the manuscript are shown in blue text. We hope the revised version meets your expectations and are happy to take further feedback.

Yours sincerely,

Elad Michael Schiller, Shilling Zhang, and Magnus Almgren

Comments from the reviewers:

Reviewer #1

This article explores how to improve the trade-off between the degree of privacy protection and the ability to use protected data. Differential privacy (DP) defines privacy protection by promising quantified indistinguishability between individuals that consent to share their privacy-sensitive information and the ones that do not. This work proposes an open-source evaluation framework for privacy protection solutions and offers evaluation for OpenDP Smartnoise, Google DP, PyTorch Opacus, Tensorflow Privacy, and Diffprivlib. In addition to studying their ability to balance the above trade-off, the authors consider discrete and continuous attributes by quantifying their performance under different data sizes. Defined criteria to quantify how different DP tools perform so that they can be selected. Evaluated and measure the impact of DP on different functionality that the studied tools provide. Two data sources of different types were used to get a nuanced picture of how well the studied tools work when using DP. The evaluation results demonstrate the degree to which the use of DP tools impacts data utility and system overhead. The choice of the proposed methods is justified.

Response: We thank the reviewer for his keen understanding and observation. We carefully choose the proposed methods in our study and spend time justifying them in the presentation so that the reader would understand the choice.

This approach is relevant. The research is of practical importance.

Response: We thank the reviewer for these words. As data and thus protected data grows, it is important to understand the toolchain, and implications thereof with tradeoffs to have useful data that is still private.

The conclusions correspond to the presented evidence and arguments, and they correspond to the basic question.

Response: Time was spent on developing the basic questions, and highlighting the conclusions for the reader of the paper.

The references appropriately. The tables, pictures and formulas meet the requirements.

Response: Again, we thank the reviewer for the comments and hence will leave references and other material as is.

Reviewer 2 Report

The manuscript entitled “Evaluation of Open-source Tools for Differential Privacy” designed a tool that can be used to evaluate Open-source Tools for DP, and showed a positive result. However, some issues still should be addressed.

1.       The author should increase the number of keywords. It is appropriate to arrange from 5 keywords to 7 keywords.

2.       In Line 57, the sentence “a randomized mechanism M is said to provide…” seems to be an inappropriate expression for academic writing.

3.       The author's introduction needs to be optimized, and we suggest that the author evaluate what needs to be improved in the introduction according to the following criteria.

·       What is the problem to be solved?

·       Are there any existing solutions?

·       Which is the best?

·       What is the main limitation of the best and existing approaches?

·       What do you hope to change or propose to make it better?

·       How is the paper structured?

4.       Kindly optimize Figure 7, the numbers are too small to see clearly.

5.       The conclusion of this paper needs to be optimized. The authors may give the details of their paper's novelty with short descriptions. It is suggested that the authors add some comparisons with previous work, advantages and disadvantages of the author's method, give some numerical results, and prospects for future research directions.

 Minor editing of the English language required

Author Response

Dear editors and reviewers,

We have carefully reviewed the comments and have revised the manuscript accordingly. Our responses are given in a point-by-point manner below. Changes to the manuscript are shown in blue text. We hope the revised version meets your expectations and are happy to take further feedback.

Yours sincerely,

Elad Michael Schiller, Shilling Zhang, and Magnus Almgren

Comments from the reviewers:

Reviewer #2:

The manuscript entitled “Evaluation of Open-source Tools for Differential Privacy” designed a tool that can be used to evaluate Open-source Tools for DP, and showed a positive result. However, some issues still should be addressed.

  1. The author should increase the number of keywords. It is appropriate to arrange from 5 keywords to 7 keywords.

Response: Thank you for this suggestion for improvement. In the revised version, we use the following keywords: Privacy protection; Differential Privacy; Open-source Tools; Evaluation; Trade-offs between privacy and utility.

  1. In Line 57, the sentence “a randomized mechanism M is said to provide…” seems to be an inappropriate expression for academic writing.

Response: We thank the reviewer for this comment. In the revised version, that sentence was rephrased. 

  1. The author's introduction needs to be optimized, and we suggest that the author evaluate what needs to be improved in the introduction according to the following criteria.
  • What is the problem to be solved?
  • Are there any existing solutions?
  • Which is the best?
  • What is the main limitation of the best and existing approaches?
  • What do you hope to change or propose to make it better?
  • How is the paper structured?

Response: We greatly appreciate the reviewer's valuable feedback on enhancing the presentation of the Introduction. We acknowledge the significance of presenting a clear and well-structured Introduction to the manuscript. In the revised version, the second paragraph of Section 1 outlines the structure of the Introduction. Specifically, we have included the definition of differential privacy (Section 1.1), a review of the studied open-source tools (Section 1.2), an explanation of our evaluation approach (Section 1.3), and a description of how our work advances the state of the art (Section 1.4). Additionally, we have added a document structure at the end of Section 1.

This structural choice was made after carefully considering the needs of the readers to facilitate their understanding of our contribution. It is important to note that our work focuses on evaluating existing solutions for machine learning and statistical queries when applying differential privacy, rather than proposing a new solution to the studied problem. We sincerely appreciate the reviewer's guidance on this matter, and we have made the necessary modifications to improve the introduction based on our thorough analysis of the reader's needs. We hope that these revisions address the reviewer's concerns, and we welcome any further feedback.

  1. Kindly optimize Figure 7, the numbers are too small to see clearly.

Response: We thank the reviewer for this comment. In the revised submission, we checked and revised all figures.

  1. The conclusion of this paper needs to be optimized. The authors may give the details of their paper's novelty with short descriptions. It is suggested that the authors add some comparisons with previous work, advantages, and disadvantages of the author's method, give some numerical results, and prospects for future research directions.

Response: We appreciate this comment. The revised version reviews the literature in a separate section (Section 4), just before the Conclusions (Section 5). Also, the Conclusions now include a discussion regarding the limitations and future work.

Reviewer 3 Report

The text is written in a logical way, presented in an interesting way. Please pay attention to the captions - figure 3-15, the letters are truncated.

Author Response

Dear editors and reviewers,

We have carefully reviewed the comments and have revised the manuscript accordingly. Our responses are given in a point-by-point manner below. Changes to the manuscript are shown in blue text. We hope the revised version meets your expectations and are happy to take further feedback.

Yours sincerely,

Elad Michael Schiller, Shilling Zhang, and Magnus Almgren

Comments from the reviewers:

Reviewer #3:

The text is written in a logical way, presented in an interesting way. Please pay attention to the captions - figure 3-15, the letters are truncated.

Response: We thank Rev.#3 for this comment. We have fixed this problem.

Reviewer 4 Report

This work developed a framework to evaluate privacy protection. The paper is well written and present relevant results. I have some suggestions for the authors, as the contributions of the work are described so far, in pages 6 and 7. It should be more detailed in the early sections in introduction.

Introduction

-          Please, review sentence in line 42.

-          Insert more information about the extraction of parameters as epsilon, delta, among others in section 1.1.

-          Please, review the sentence ” Machine learning tasks deal with the construction of models based on training data from a given dataset, say, for linear regression”.

-          Provide more information about ML and statistical approach, as their advantages and disadvantages for DP problem.

-          Table 1 should be in journal’s template. I do not see any necessity of Table 1 because the author insert this information in text.

-           

Evaluation Settings

-          Why other databases were not applied?

-          Were some methods to evaluate the relevant statistical parameters were applied?

-           

Evaluation results

-          Figure 3 has some labels that are not visible.

-          The indices from (a) to (p) should be inserted in Figure legend too.

-          As presented in Figure 3, 4, and 5, the epsilon value have some importance for little numbers. How it was explored?

-          Although the authors comment that the proposed analysis was done for the first time, the discussion with related works or that have the same proposal should be done at the end of section 3. It is necessary that there be this comparison.

Author Response

Dear editors and reviewers,

We have carefully reviewed the comments and have revised the manuscript accordingly. Our responses are given in a point-by-point manner below. Changes to the manuscript are shown in blue text. We hope the revised version meets your expectations and are happy to take further feedback.

Yours sincerely,

Elad Michael Schiller, Shilling Zhang, and Magnus Almgren

Comments from the reviewers:

Reviewer #4:

 I have some suggestions for the authors, as the contributions of the work are described so far, in pages 6 and 7. It should be more detailed in the early sections in introduction.

Response: We thank Rev. #4 for this comment. In the revised version, paragraph 2 of section 1 summarizes the paper's contribution and explains the structure of section 1. Also, we have shortened Section 1, and now the detailed version of our contribution starts on Page 5 rather than Page 6.

Introduction

-          Please, review sentence in line 42.

Response: We appreciate this comment. In the revised version, we have revised the sentence and explained that privacy-sensitive information cannot be revealed effectively.

-          Insert more information about the extraction of parameters as epsilon, delta, among others in section 1.1.

Response: We thank the reviewer for the comment. In the revised version, we are presenting the basic definition of differential privacy, which uses only one parameter, ε. While the parameters in differential privacy can vary, they are determined by practitioners on their use case, i.e., whether they need a higher level of privacy or more accuracy in data analysis. Thus, in the sequel, we explain the role of such parameter, δ.

-          Please, review the sentence ” Machine learning tasks deal with the construction of models based on training data from a given dataset, say, for linear regression”.

Response: We thank Rev. #4 for this comment. In the revised version, we have clarified that linear regression is one of the effective ways in which a model can be constructed.

-          Provide more information about ML and statistical approach, as their advantages and disadvantages for DP problem.

Response: We appreciate this suggestion for improvement. In the revised version, we have clarified that our work aims to evaluate the performance of ML and statistical approaches when applying DP. We hope that now the text is clearer to the reader, and we welcome further feedback.

-          Table 1 should be in journal’s template. I do not see any necessity of Table 1 because the author insert this information in text.

Response: We welcome this comment from Rev. #4. In the revised version, for the sake of compliance, we have removed Table 1.

Evaluation Settings

-          Why other databases were not applied?

Response: We thank Rev. #4 for the comment. The revised version explains that indeed there are a number of datasets available. However, we have to choose the datasets that can (i) fit the functionalities the investigated tools can provide, and (ii) have diversified data settings so as to better examine the performance of the tools. The studied datasets were selected since they can both be applied to statistical analysis and machine learning tasks which are covered by the considered tools. Furthermore, one of our selected datasets, i.e., the health survey data, contains integer data that represents categorization, while the other dataset uses continuous data. Such diversity is what we looked for as it can offer a nuanced picture of how the considered tools work with different types of data.

-          Were some methods to evaluate the relevant statistical parameters were applied?

Response: We would like to thank Rev. #4 for this comment. In the revised version, we explain the fact that previous literature focused on exploring differential privacy in statistical analysis with smaller or different sets of query functions; see the end of Section 1.4.1.

Evaluation results

-          Figure 3 has some labels that are not visible.

Response: We thank the reviewer for this comment. We have fixed this problem.

-          The indices from (a) to (p) should be inserted in Figure legend too.

Response: We thank the reviewer for this comment. We have fixed this problem.

-          As presented in Figure 3, 4, and 5, the epsilon value have some importance for little numbers. How it was explored?

Response: We appreciate this comment from Rev. #4. In the revised version, we further help the reader to understand the relationship between the epsilon values and the utility degradation, see Section 3.1.1 (just before Section 3.1.2).

-        Although the authors comment that the proposed analysis was done for the first time, the discussion with related works or that have the same proposal should be done at the end of section 3. It is necessary that there be this comparison.

Response: We appreciate Rev. #4's comment. In order to address this concern and acknowledge the relevant works in the field, we have added a new section that specifically compares our results with earlier studies, providing a comprehensive discussion and fulfilling the request for comparative analysis. Additionally, we have included new references to relevant literature and emphasized the novelty of our work. This new section appears just after Section 3 and before the Conclusions (Section 5) to ensure its proper placement.

Reviewer 5 Report

This manuscript evaluates different open-source tools for implementing differential privacy. The authors claim that they provide an evaluation framework. Overall, the manuscript is well written. I can access or understand the motivation, methodology, and results of this paper clearly.

Here are some of my suggestions.

1. Does a framework for evaluating different differential privacy tools exist in prior work? What ideas do the authors' proposed methods draw on?

2. The punctuation after Equation 1 needs attention.

3. The evaluation results and discussions are written in great detail, which is appreciated. However, I would like to know if, the authors have identified any shortcomings of the proposed framework and further ideas for improvement on the basis of these results.

4. Authors are suggested to add the prospect of future work in the conclusion section.

Author Response

Dear editors and reviewers,

We have carefully reviewed the comments and have revised the manuscript accordingly. Our responses are given in a point-by-point manner below. Changes to the manuscript are shown in blue text. We hope the revised version meets your expectations and are happy to take further feedback.

Yours sincerely,

Elad Michael Schiller, Shilling Zhang, and Magnus Almgren

Comments from the reviewers:

Reviewer #5;

  1. Does a framework for evaluating different differential privacy tools exist in prior work? What ideas do the authors' proposed methods draw on?

Response: We thank Rev. #5 for this question. To the best of our knowledge, such a framework is not offered as an open-source tool, but there are earlier evaluations of DP. In the revised version, we state that Garrido et al. [12] compared the performance of Google DP, SmartNoise, diffprivlib, diffpriv, and Chorus with respect to the features offered to the users. However, Garrido et al. do not consider issues related to machine learning, as we do. Also, Hay et al. [29] proposed DPBench, which is an evaluation framework for privacy-preserving algorithms but only in the context of histograms. More details appear in Section 4.

Regarding the ideas behind our evaluation method. In the revised version, we explain that this work has taken a black-box approach to evaluating the studied DP tools.

  1. The punctuation after Equation 1 needs attention.

Response: Fixed.

  1. The evaluation results and discussions are written in great detail, which is appreciated. However, I would like to know if, the authors have identified any shortcomings of the proposed framework and further ideas for improvement on the basis of these results.

Response: We are thankful for this question. In the revised version, we have explained the limitations of our proposal in Section 5, just before future work.

Specifically, the studied approach did not include the conduction of a white box evaluation of the DP tools since we wished to emphasize the most typical use case scenarios. Furthermore, in our current work, we have not explicitly considered the correctness aspect of the DP tools. While our evaluation focused on performance and utility, it is essential to assess the correctness of the DP mechanisms to ensure that the privacy guarantees are effectively preserved. Such assessment requires another set of analytical tools and cannot be achieved using only empirical results.

  1. Authors are suggested to add the prospect of future work in the conclusion section.

Response: We appreciate this suggestion. In the revised submission, we offer the reader directions for future work on our open-source framework. As an open-source solution, it can serve other researchers as an evaluation framework for future DP tools, see Section 5 for details.
